# 'ARVs are a constant reminder of lost freedom, whereas for others, they are liberating': understanding the treatment narrative among people living with HIV (PLHIV) in Malawi and Zimbabwe–a qualitative study

Sehlulekile Gumede-Moyo [1,2] Sunny Sharma,[3] Clorata Gwanzura,[4] Rose Nyirenda,[5] Philip Mkandawire,[6] Kumbirai Chatora,[7] Nina Hasen[8]

For numbered affiliations see end of article.

**Correspondence to**
Dr Sehlulekile Gumede-Moyo; seh.sokhela@gmail.com

## ABSTRACT

**Objective** The aim of the research was to understand what drives and motivates young women living with HIV in their treatment journeys, as well as their key influencers. The findings will assist in appreciating their obstacles to treatment and constructing the most effective ways to convey fresh messages to them. These insights will contribute to the messaging developed for a campaign across sub-Saharan Africa, primarily Malawi and Zimbabwe.

**Design** This was a qualitative study conducted in order to build an understanding of unknown parts of the HIV treatment journey through in-depth interviews. A hybrid approach was used to conduct thematic analysis.

**Setting and participants** Study participants were HIV-positive women, their influencers (eg, parents/partners/siblings/aunts and uncles/religious leaders, etc) and healthcare providers from a range of regions in Zimbabwe and Malawi.

**Results** A total of 26 young people living with HIV (PLHIV), 29 healthcare providers and 24 influencers such as maternal figures in the community were interviewed. Two main broad insights were drawn, that is, key needs and wider contextual factors. The key needs of young PLHIV to stay on treatment were empathy, support and emotional connection with antiretroviral therapy (ART), while the wider contextual factors impacting their treatment journeys were structural challenges and cultural reference points. Fear of shame and humiliation can also be significant barriers to disclosure and treatment. The fear predisposes the PLHIV to the need for empathy, support and connection with ART. Mental health and anxiety appear to be comorbid with HIV. Some PLHIV have a small and limited support network leading to very few people living openly with HIV. There is no 'one-size-fits-all' approach, and maximising campaign reach will likely need a multifaceted approach.

**Conclusion** Currently, the relationship between nurses and PLHIV can appear to be transactional. Through learning about the community (including PLHIV), there are more chances of communicating in a way that resonates.

## STRENGTHS AND LIMITATIONS OF THIS STUDY

⇒ The participants of the study were sampled from a range of people living with HIV (PLHIV), their influencers and healthcare providers.
⇒ The depth of inquiry allowed to identify potential areas of interest which were literacy and awareness of treatment.
⇒ Several campaigns reinforcing how people 'should' feel about PLHIV may have influenced the research participants to respond in a social desirable manner.

## INTRODUCTION

The Joint United Nations Programme on HIV/AIDS (UNAIDS) 95-95-95 targets aim to achieve a 95% level of diagnosis of people living with HIV (PLHIV), 95% of those diagnosed on antiretroviral therapy (ART) and 95% of those on ART virally suppressed by 2025.[1–4] In 2020, of all PLHIV globally, 84% knew their status, 73% (57–88%) were accessing treatment and 66% (53–79%) were virally suppressed.[5] East and Southern Africa, the regions most affected by HIV, are home to the largest numbers of PLHIV.[6] While the number of PLHIV in East and Southern Africa continues to increase, access to ART is increasing as well.[7] Young people, especially women and key populations, continue to have a disproportionately low level of viral suppression.[4] One goal of HIV care and treatment is achieving an undetectable viral load as it confers benefits at both the individual and public health level.[8 9] There is strong evidence that supports that PLHIV cannot sexually transmit HIV to others when their viral load is suppressed to 200 HIV RNA

copies/mL or lower, and they have been adherent to ART for at least 6 months.[8 10–12]

UNAIDS has called for combined approaches to scale up HIV prevention, to reinvigorate the global response and make a sustained impact on global HIV incidence rates.[13] Combining separate strategies for different indications into singular prevention products can reduce the stigma around HIV and sexually transmitted infection, improve acceptability of and adherence to more convenient products, and be more cost-effective by addressing overlapping risks.[14 15] All prevention programmes require a strong community empowerment and specific efforts to address legal and policy barriers, as well as the strengthening of health and social protection systems, plus actions to address gender inequality, stigma and discrimination.[16 17]

The majority of the existing HIV treatment campaigns in sub-Saharan Africa (SSA) are one dimensional when it comes to the benefits of treatment, as treatment extends life, being the key benefit, rather than focusing on other factors which could resonate more personally with each PLHIV. The aim of the research was to understand what drives and motivates young women living with HIV, as well as their key influencers. The findings will assist in appreciating their obstacles to treatment and constructing the most effective ways to convey fresh messages to them.

## METHODS
### Study design
This was a qualitative study conducted in order to build an understanding of unknown parts of the HIV treatment journey through in-depth interviews with healthcare workers, young PLHIV and their influencers (eg, parents/partners/siblings/aunts and uncles/religious leaders, etc). The potential areas of interest were literacy and awareness of treatment, understanding their treatment journey (what is communicated to them about treatment, how well is it understood, what is compelling (if anything), barriers) and ideas around bringing a community approach to supporting PLHIV with treatment.

### Study setting
The research seeks to deliver foundational insights on the current state of awareness of viral suppression and treatment as prevention, including the awareness and motivations of PLHIV, their influencers and healthcare providers (HCPs) to stick with, counsel or support taking their medicine every day. These insights will contribute to the messaging developed for a campaign across SSA, primarily Malawi and Zimbabwe, that will aim to shift perspectives of PLHIV to take action based on the new concept/narrative of treatment literacy.

### Sample recruitment
Our main study participants were HIV-positive women, their influencers and HCPs from a range of regions in Zimbabwe and Malawi. The breakdown of study participants is indicated in online supplemental table 1. Eligibility criteria and recruitment procedures are described below. Young HIV-positive women (18–35 years) were targeted as they have disproportionately low levels of viral suppression based on UNAIDS reports.[4–6]

### People living with HIV
The participants' names were retrieved from the Population Services International-assisted ART clinic's database. The eligibility for the PLHIV was for one to be: 18–35 years old (diagnosed at least 12 months ago) and not pregnant or breast feeding. Quotas were set to recruit one-third of participants on treatment with good adherence and two-thirds not taking treatment (non-adherence)—a mix of respondents who had been on treatment for long and shorter periods of time, and a mix of those who were virally suppressed and those who were not.

### Healthcare providers
Eligibility criteria for HCP recruitment included having at least 3 years' experience in their current role, spending at least 50% of their time working in HIV care and interacting with at least five HIV-positive women (aged 18–35 years) each day. HCPs worked at a variety of sites including public and private facilities.

### Influencers
For this study, an influencer was defined as someone who has an impact on PLHIV, either as part of their treatment journey or friends/relatives. The influencers were recruited after initial analysis of the PLHIV transcripts which determined who they were. These were maternal/grandmother figures, workshop leaders and WhatsApp group leaders. Workshops are support groups where PLHIV can meet to discuss concerns, ask questions and receive peer support; workshop trainers are individuals who run these workshops. WhatsApp group chats are set up among communities to act as peer support networks consisting of PLHIV and their family/friends. WhatsApp group leaders are individuals who run these group chats in the communities. These influencers were recruited through snowballing.

### Data collection
Data were collected from December 2020 to February 2021 in both countries. Semistructured interviews were conducted by trained moderators with vast experience in qualitative data collection. Guides were written in English, and then translated to Chichewa for Malawi, and into Ndebele and Shona for Zimbabwe. Study participants were asked before beginning the interview their language preference. Pretesting of the guides was done according to procedure. Due to the developing COVID-19 pandemic, planned face-to-face interviews were replaced by in-depth telephone interviews lasting approximately 45–75 min. If at any point phone lines were interrupted or participants had to pause/stop the interview, the interviewers arranged to call back at a suitable time. All interviews were audio-recorded for translation and transcription.

Each participant was allocated with a unique personal identifier to maintain anonymity.

## Data analysis

Smart transcript was used to group the responses in Excel. A hybrid (ie, inductive and deductive) approach was used to conduct thematic analysis. Four research analysts (two per country) independently coded the transcripts, after familiarising themselves with the scripts. The coded data were then shared with field managers who had field notes as additional sources of data. In cases where there were discrepancies, all coders rechecked the transcripts and held meetings with the principal investigators and research managers until agreement was achieved. Anonymised quotes from some participants were also included in the manuscript to illustrate certain themes.

## Patient and public involvement

The study participants were involved in the research recruitment, as they were the ones who led the identification of influencers; however, they were not directly involved in the design and conception of this study.

## RESULTS

Two main broad insights were drawn from thematic analysis, that is, key needs and wider contextual factors. The key needs of young PLHIV are empathy, support and connection with ART, while the wider contextual factors with impact on their treatment journeys are structural challenges and cultural hooks.

## Key needs

Fear of shame and humiliation can be significant barriers to disclosure and treatment. The fear predisposes the PLHIV to the need for empathy, support and connection with ART.

### Empathy

There was a disconnect between how HCPs and PLHIV view the perceptions of the disease in the community, which can act as a barrier to connection between the messenger and the patient. HCPs are an important messenger; however, there can sometimes be a gap in understanding or lack of connection which can mean their messages are less credible to PLHIV.

> Life here is a bit difficult if people discover that you take medicine they either discriminate you or talk behind your back, so we don't disclose to people. (PLHIV, Malawi)

> Gone are the days when it was a taboo to talk about HIV but nowadays it's now common and people accept all the messages they hear about HIV however it differs from culture to culture but generally people have accepted that HIV exists. (ART nurse and midwife, Zimbabwe)

Although HCPs say they treat PLHIV like 'anybody else', some PLHIV face discrimination in the community. For some, HIV status is not openly disclosed to the wider community as it can impact relationships.

> When it comes to professionals I guess they keep and maintain work ethics and professionalism, they treat us like any other person, they are supportive and encouraging compared to community at large. (PLHIV, Zimbabwe)

PLHIV often fear the worst and they often worry more about 'what will people think?' rather than 'what can I do to get better?' PLHIV expressed fear of judgement when telling a partner in case they were not accepting of their status, and not wanting to be seen taking medication and fear of rejection by the community.

Most of the PLHIV were more fearful of judgement as they do not want to be seen as promiscuous.

> People are so judgemental to the extent that they would say she got it through promiscuity. (PLHIV, Zimbabwe)

Initiating treatment at diagnosis is not always optimal because PLHIV are catastrophising, while nurses are providing a rational 'treatment seminar'. Test and treat means that most patients are initiated to treatment straight away, but this is often not the best time to learn about treatment benefits and the importance of adherence as a patient's mental health (shock, numbness, fear, 'fight or flight', denial) makes it difficult to focus on information.

> It gave me a lot of stress, I didn't have peace and I was scared, should I tell my mom, should I start medication or I shouldn't start because of the rumours that the drugs deform people so I used to have a lot of doubts and it took me time to accept it, although at the hospital they said they weren't going to give me a chance to think about it, they said I had to start the very same day. (PLHIV, Malawi)

Taking treatment is a sign of HIV-positive status so PLHIV need to hide their medication; however, treatment can also be a reliever enabling the patient to show no signs of infection. PLHIV do not want members of the community to know their status, so they lead a 'double life' to prevent discrimination.

> I feel like I am not different from others because I have no signs that shows that I am infected with the virus. At the same time this virus distanced me from my friends that I used to associate with. It changed my entire lifestyle because at the present moment I am living in a lie because I don't want people to know that I am infected. (PLHIV, Zimbabwe)

Religious influence plays a major role in people's lives and the perception of PLHIV within the community. Some religious communities are reluctant to talk about HIV which may contribute to preconceptions towards

the disease and further prevent disclosure or adherence. However, as HIV is not regularly spoken about in church, PLHIV fear that they will not receive the empathy they need in the church to pull back from the catastrophe, nor does the church help overcome a major hurdle, while spreading the message that all people including those with HIV are worthy of love equally with God.

> Many people believe that when one is HIV positive that person was very promiscuous, that is why people prefer to make it a secret about their status they do not want to be judged. Even in the community and some church organizations they do not want to talk about the virus. (ART nurse, Zimbabwe)

For some, antiretrovirals are a constant reminder of lost freedom, whereas for others, it is liberating.

> I always think that if not for this disease, I could not have been taking this medicine and I could have been a free person. (PLHIV, Malawi)

> Living with HIV doesn't mean you are dying or you are disabled it's just a condition which you can live with, and doesn't affect your lifestyle as long you take treatment as prescribed. (PLHIV, Zimbabwe)

### Support

Often PLHIV are in shock when they receive their diagnosis, and the patients need support to accept their status to begin treatment.

> When one gets tested they are given some shallow counselling which is superficial, and start treatment, the treatment becomes very difficult because the person will not have understood, and they are still in shock maybe because they were not expecting that result. (ART nurse and midwife, Zimbabwe)

Helping people accept their status is a key challenge; the sooner people accept their status, the more likely they are to adhere and move on with their life.

> I was worried because what I did not expect in my life has happened. But through meeting my good friends with good counselling things started changing. (PLHIV, Malawi)

Many are concerned about disclosing their illness to their partner and family, but when they have a positive response (most do), this makes living with the disease much easier as they have support from people around them.

> I want to thank my young sister for support and encouragement she gave me, as young as she was, she was my pillar of strength even at a time when I wanted to quit, stood with me through out. (PLHIV, Zimbabwe)

Depression and anxiety can be a significant disruptor, but there is little mental health support. Treating the PLHIV holistically beyond suppressing the HIV alone is important to maintain adherence.

> I was depressed because of my HIV status, it stressed me a lot. Even now when I think about it deeply, it still stresses me. (PLHIV, Zimbabwe)

### Connection with ART

HCPs felt that there was good treatment literacy in both Zimbabwe and Malawi. They demonstrated significant HIV knowledge and were confident communicating messages to patients. Gaps noted by HCPs were that some PLHIV struggling with adherence tended to do so because they did not understand disease severity or benefits of medication. Knowledge of viral suppression varies across countries, but it is a motivating message. However, there is a good basic level of knowledge around treatment benefits across all PLHIV.

HCPs in Zimbabwe did not widely communicate information about viral suppression. PLHIV clearly understood the benefits of treatment and what motivated them to adhere (being healthy, controlling the virus, living a normal life, not looking like they have HIV to the community), but there were very few who mentioned viral suppression or reduction in viral load.

In Malawi, some HCPs communicated information about viral suppression to PLHIV, and PLHIV had a reasonable understanding of the concept. They talked about the fact that they could reach a point where the virus is not evident in their bloodstream and know that this means they will not transfer the virus to their sexual partner or child.

HCPs saw the effectiveness and impact that understanding viral suppression could have on driving behaviour change and adherence to treatment, especially when PLHIV are able to link viral suppression to feeling better, and become more motivated to adhere to treatment.

> They get motivated…if we tell a client that your levels are undetectable and if you continue taking medication like that you will have a long time before getting sick. Once a client hears that, they get motivated and make promises that they will maintain taking medication properly. (clinical officer, Malawi)

### Contextual factors
#### Structural challenges

Poverty-related challenges remain a central barrier to adherence, and the COVID-19 pandemic made the situation worse (figure 1). PLHIV cannot take their medication if they do not have enough food to eat, even if they are motivated to do so. Sourcing food becomes core priority and HIV care is less prioritised. COVID-19 has exacerbated existing financial challenges.

> These people must be a priority because it can be anyone, your mother, brother, sister or friend. These

**Food scarcity**

PLHIV cannot take their medication if they do not have enough food to eat, even if they are motivated to do so. Sourcing food becomes core priority and HIV care is down-prioritised. COVID-19 has exacerbated existing financial challenges.

**Access to healthcare**

Not being able to afford transport to the clinic means that some PLHIV miss appointments and do not collect treatment. This is a particular challenge in rural areas, and mentioned more in Malawi.

**Overtly technical language**

Information from HCP may not be understood, unless explained in terms which are relatable. Some PLHIV struggle to read information leaflets, posters, and other written communication about HIV because they are too technical.

**Lack of resources at clinics**

Not enough staff meaning that less time is spent educating patients. Resource challenges in Malawi include electricity blackouts, inconsistent condom supply, occasional low stock of medications.

**Lack of access to digital channels**

Many PLHIV do not have internet access. Some people using older models of mobile phone. This is important when considering campaign channels.

**Figure 1** Structural challenges. HCP, healthcare provider; PLHIV, people living with HIV.

people need care so that they do not affect others. During budget allocation the government must prioritise the Health sector and avail funds towards their welfare and resources as first preferences. I believe everyone has a role to play in the care of PLHIV. (influencer, maternal figure, Zimbabwe)

Some PLHIV were not able to afford transport to the clinics leading to missed appointments, which affected treatment schedule. This was a particular challenge in rural areas and mentioned more in Malawi.

It is difficult when patients miss appointments and don't take their medication as their viral load gets higher and you have to spend more time educating them about taking treatment properly. (client officer, Malawi)

There is also a lack of resources in health facilities. Most health facilities do not have enough staff, meaning that less time is spent educating patients. Resource challenges include electricity blackouts, inconsistent condom supply and occasional low stock of medications.

It's a challenge because sometimes the ART may not be available, sometimes even the testing kits are not available, and samples are failing to go to the lab on time, end up getting stale due to lack of resources. Initially, we are supposed to retest and do CD4 counts tests every now and then to know the patient's viral load, but unfortunately we are failing to do that. We watch people deteriorate because we have nothing to help them with. (ART nurse, Zimbabwe)

I could have recommended that they give us more nurses so that people get helped faster. (ART nurse, Malawi)

PLHIV stated that because of overly technical language, information from HCPs may not be understood, unless explained in terms which are understood. Some PLHIV struggle to read information leaflets, posters and other written communication about HIV because they are too technical.

### Cultural hooks

People in Malawi and Zimbabwe are diverse with different hobbies and interests. There is no 'one-size-fits-all' approach, and maximising campaign reach will likely need a multifaceted approach.

Young PLHIV may struggle with their disease but still find ways to have fun and live life.

They have identified female role models who represent independence, open-mindedness and progress. There is a need to show PLHIV that they too can be educated, independent and valued members of society—just like their role models. These role models embody PLHIV's aspirations and dreams, which are focused on 'thriving'—becoming independent, providers, and being respected by their families and communities.

I think this must be done by everyone because people trust different people, it should be an issue discussed at household level, community, all HCPs, NGOs [non-governmental organisations] and social media. (influencer, trainer, Zimbabwe)

Some people think HIV ended and are becoming careless. So it's important that we are not quiet, we have to do awareness campaigns, encouraging people, reminding them about HIV and its dangers. (influencer, religious leader, Malawi)

## DISCUSSION

This study provided some foundational insights into the current state of awareness of viral suppression and treatment as prevention, including the awareness and motivations of PLHIV, their influencers and HCPs. The key themes demonstrated that there is a need for empathy and for attention to be paid to wider contextual factors in communication campaigns.

Early life experiences of HIV are scary and/or traumatic. Associations include loved ones being 'taken away' and not returning, seeing people in the community who are sick and frail while people say they have AIDS as a catch-all for any serious illness.[18 19] For healthcare workers, specifically ART nurses, treating HIV might feel like a routine, but for every PLHIV, it represents a profound change in their lives often characterised by fear, shock, sorrow, feeling of loss and low self-worth.[20 21] Our study revealed a disconnect between how HCPs and PLHIV view the perceptions of the disease in the community, which can act as a barrier towards connection between the messenger and the patient. Focusing too heavily on educating and getting PLHIV onto treatment can mean that HCPs fail to meet the emotional needs of PLHIV, leaving them feeling even more overwhelmed and deflated.

There is a need to create an opportunity for the HCP–PLHIV interaction to become more transformational, rather than transactional. HCPs tend to be focused on educating to make sure PLHIV are well informed on HIV, encouraging disclosure, tracing contacts and encouraging adherence to treatment. HCPs see their role primarily as educators, so listening comes second. On the other hand, PLHIV have a lot to process in terms of diagnosis: what implications this has in their life and those around them and the new information on HIV and care.

HCPs should be orientated on the fact that rationality regarding treatment-taking extends beyond the biomedical realm, requiring adjustments to sense of self and identity, and decision-making that is situated and socially embedded.[21–23] However, in many clinical centres, standards and guidelines surrounding HIV treatment communication may be absent or lack specificity, which is a hindrance.[23] HCPs can be trained to understand the types of support different patients need at different points of time. Treatment literacy campaigns should also consider how to still resonate with people when they are going through tough times.

It is important to understand that some PLHIV 'catastrophise' every potential disclosure of their status, particularly when out of their control. In times of hardship, people in Malawi and Zimbabwe turn to the church as a message of hope and comfort for a better future. HIV is a traumatic experience for PLHIV and they can feel like they have committed a sin. As HIV is not spoken about at a community level, they are scared that they will be shunned by all.

HCPs are already using a range of communication tools to convey viral suppression and treatment benefits in an accessible way. This was mirrored by PLHIV when expressing their knowledge of HIV. These tools could be used to scale up the viral suppression message and reach a point where it is well recognised and understood by not just PLHIV, but the wider community too. Studies conducted in South Africa, Malawi and Kenya among health providers reported mixed attitudes about treatment as prevention, although most were supportive.[24–27] In Malawi, HIV care providers and programme stakeholders expressed concerns related to equating 'undetectable' with 'healed', which may impact adherence negatively, and to a potential increase in promiscuity and HIV reinfection.[25] Kenyan health providers reported fears that telling patients undetectable viral loads will lead to other risk behaviours, or that consequent HIV transmission would be blamed on them.[26]

Busy clinics can mean less time spent with PLHIV and can result in gaps in education. Patients who are defaulting tend to do so because they do not understand disease severity or benefits of medication. PLHIV may still be in denial stage at the time of diagnosis and treatment. Patients are not in the right frame of mind to absorb key messages about benefits and the importance of adherence at initial treatment stage. Our findings confirm the health systems barriers such as limited and unprofessional providers at healthcare facilities, unavailability of drugs and long travel distance that have been identified by other researchers.[23 28–31]

### Strengths and limitations of the study

We obtained a rich and nuanced appreciation of the current state of awareness of viral suppression and treatment as prevention, including the awareness and motivations as we sampled from a range of PLHIV, their influencers and HCPs. The depth of inquiry allowed us to identify potential areas of interest, which were literacy and awareness of treatment, treatment journey and ideas around bringing a community approach to supporting PLHIV with treatment. These issues were also considered in depth through a unique lens of influencers which were sampled from a variety of positions within the communities to allow a broad review.

Caution should be exercised when interpreting the results of this exploratory analysis due to small sample size and limited breadth of influencers sampled. Several campaigns have reinforced how people 'should' feel about PLHIV; hence, HCPs and influencers may have answered questions based on what they think is 'correct' or 'acceptable', rather than their true thoughts or beliefs which may be undesirable. The insights of the study are likely to contribute to the development of campaigns across SSA, and

creating of messages that can go beyond the medical and PLHIV living a longer life.

## CONCLUSION

Treatment literacy campaigns should also consider how to still resonate with people when they are going through tough times. Currently, the relationship between nurses and PLHIV appears to be transactional. Some HCPs are hesitant to speak about viral suppression denoting the end of onward transmission risk. There is a need to show PLHIV that they too can be independent and valued members of society. Through learning about the community (including PLHIV), there are more chances of communicating in a way that resonates.

**Author affiliations**
[1]Healthcare, Ipsos MORI UK, London, UK
[2]Epidemiology and Population Health, London School of Hygiene & Tropical Medicine, London, UK
[3]Ipsos MORI UK, London, UK
[4]Ministry of Health and Child Care, Harare, Zimbabwe
[5]Ministry of Health and Population Malawi, Lilongwe, Malawi
[6]Population Services International, Lilongwe, Malawi
[7]Population Services International, Harare, Zambia
[8]HIV and TB, Population Services International, Washington, District of Columbia, USA

**Acknowledgements** This work was supported with leadership by the Ministry of Health and Population in Malawi and the Ministry of Health and Child Care (MoHCC) in Zimbabwe. We thank our partners, Population Services International in Malawi and Zimbabwe, for collaborating on the implementation of this research.

**Contributors** SG-M and SS were responsible for conceptualisation, formal analysis, supervision, funding acquisition, validation, investigation and writing the original draft. CG and RN contributed to resource mobilisation, investigation and project administration. PM, KC and NH contributed to the conceptualisation, resource mobilisation, funding acquisition and project administration. SG-M is responsible for the overall content as the guarantor. All authors read and approved the final manuscript.

**Funding** This study was supported by the Flip the Script Project funded by Johnson & Johnson and Bill & Melinda Gates Foundation.

**Competing interests** None declared.

**Patient and public involvement** Patients and/or the public were involved in the design, or conduct, or reporting, or dissemination plans of this research. Refer to the Methods section for further details.

**Patient consent for publication** Consent obtained directly from patient(s).

**Ethics approval** This study involves human participants and was approved by the College of Medicine Research & Ethics Committee (COMREC) and Medical Research Council of Zimbabwe (MRCZ/A/2683). Verbal explanation of the research purpose was given prior to conducting the interviews. All participants gave verbal consent over the phone for the recording and transcription of interviews which was documented in an informed consent form.

**Provenance and peer review** Not commissioned; externally peer reviewed.

**Data availability statement** Data are available upon reasonable request. This study used data obtained from human participants. The dataset (anonymised survey responses) is owned by the governments of Malawi and Zimbabwe.

**ORCID iD**
Sehlulekile Gumede-Moyo http://orcid.org/0000-0003-1480-6534

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
