## [Reviewer comments · BMJ Open]

This paper was submitted to a another journal from BMJ but declined for publication following peer review. The authors addressed the reviewers' comments and submitted the revised paper to BMJ Open. The paper was subsequently accepted for publication at BMJ Open.

ARTICLE DETAILS

TITLE (PROVISIONAL)	"ARVs are a constant reminder of lost freedom whereas for others they are liberating": Understanding the Treatment Narrative among People Living with HIV (PLHIV) in Malawi and Zimbabwe - A qualitative study
AUTHORS	Gumede-Moyo, Sehlulekile; Sharma, Sunny; Gwanzura, Clorata; Nyirenda, Rose; Mkandawire, Philip; Chatora, Kumbirai; Hasen, Nina

VERSION 1 – REVIEW

REVIEWER	Sineke, Tembeka University of the Witwatersrand School of Public Health, Health Economics and Epidemiology Research Office
REVIEW RETURNED	05-Aug-2022

GENERAL COMMENTS	Abstract: The objective is lengthy and all over the place. Rewrite a clear and concise aim followed by the objectives. It would be useful to include additional demographics such as age etc. The results seem to only include PLHIV perspective, its difficult to tease out the influencers or health provider perspectives. Methods: - Is there a specific reason why you only included women? - Details about the questions included in the interview guide are unclear. Results: The results are not presented - I am not sure if empathy is the correct theme to summarise the results under 3.1.1. You need to reorganize your sub-themes to tell a story. The summary text does not fully describe the quotes. Consider moving the "support" theme to the beginning of the results section. 3.1.3 Has lots of summaries but no quotes to support your statements. Discussion: The discussion is well written however, it is difficult to appraise considering that the results not well organized and do tell the full story. References: References need to be formatted -Add hyperlinks for press releases and online articles.
---

REVIEWER	Carlander, Christina Karolinska University Hospital
REVIEW RETURNED	08-Nov-2022

GENERAL COMMENTS	This qualitative study performed in Zimbabwe and Malawi sought to deliver insights on the current state of awareness of viral suppression and treatment as prevention among PLHIV, influencers and healthcare providers. The study was well executed. The paper is well-written and easy to comprehend. It was interesting that so much was similar to the feelings/experiences that my patients (of whom a majority are migrants from sub-Saharan Africa) are sharing with me and co-workers. I have some comments.  • In the Introduction section (page 4, lines 47-57) four questions are posed which the study aimed to answer regarding awareness of treatment and viral suppression. Even though I appreciate that focus should be on the insights brought from the thematic analysis I would consider adding to the Discussion a reflection on how well the study managed to answer the questions posed in the Introduction. • How was the sample size decided upon? Was sampling done until redundancy in data was reached (as common in qualitative research)? • Consider including a little more data on PLHIV in Table 1. Time since HIV diagnosis? Time since start of ART? This would make it easier to assess transferability.
---

REVIEWER	Stroumpouki, Theodora Kingston University
REVIEW RETURNED	14-Nov-2022

GENERAL COMMENTS	Dear authors, Thank you for submitting a manuscript on the very important topic on the current state of awareness of viral suppression and treatment for PLHIV, their influencers, and healthcare providers. Please, consider the following comments constructively so that you can improve the content of this research paper. The abstract sets the scene and gives to the reader a clear idea of the content of the paper. 'Design' and 'Setting and Participants' should be included in a broader section as 'Methodology'. The method of data analysis used should be also stated in the Abstract, prior to 'Results'. Introduction section: Could you please include a clear aim linked to the research questions? Methodology section: Could you please specify the type of qualitative design (line 7)? Is the design based on a specific theoretical framework? In section 2.3, could you please justify the reason you included only women? Is this due to gender inequality issue? If yes, could you please explain more and include evidence in the Introduction section that can provide the rationale for your choice? Could you please explain how trustworthiness has been achieved? Also, did the participants confirm the analysed results? How did this confirmation process occur? The data analysis section provides details on the process used to analyse the data. Was this process based on a specific model/step process?
--

	Ethical clearance: Were the participants able to withdraw from the study? How was confidentiality and anonymity maintained? Results and Discussion were well presented. Findings are discussed in relation to the original research questions and sub-themes were well supported by narratives. Discussion includes secondary evidence that supports the findings. Citation 20 and 24 though are very old. Is there any specific reason for that? Could you find more current evidence? Strengths and Limitations: Do you think that the non-participation of male PLHIV could reveal different results? Could this be a limitation, too? Conclusions were well considered and some recommendations for further practice were provided. The topic generates new knowledge on the essential components of HIV prevention, such as HIV education and awareness. Well done!
--	---

VERSION 1 – AUTHOR RESPONSE

Reviewer Comments: 1

Miss Tembeka Sineke, University of the Witwatersrand School of Public Health

Comments to the Author:

Abstract: The objective is lengthy and all over the place. Rewrite a clear and concise aim followed by the objectives. It would be useful to include additional demographics such as age etc. The results seem to only include PLHIV perspective, it's difficult to tease out the influencers or health provider perspectives.

Response:

*The objective of the study has been reconstructed as recommended. **Page 4, lines 28-31** and in the abstract as well.*

Methods:

Is there a specific reason why you only included women? - Details about the questions included in the interview guide are unclear.

Response:

The study sample only included women living with HIV as these have disproportionately low levels of viral suppression based on UNAIDS reports. Literature with regards has also been included in the

introduction. Page 4, line 8-9.

Results: The results are not presented

- I am not sure if empathy is the correct theme to summarise the results

under 3.1.1. You need to reorganize your sub-themes to tell a story. The summary text does not fully describe the quotes. Consider moving the "support" theme to the beginning of the results section.

3.1.3 Has lots of summaries but no quotes to support your statements.

Response:

Thank you for your contribution, however we are confident that the key needs identified in this study are empathy, support and connection to ART in that order based on our thorough analysis.

Discussion: The discussion is well written however, it is difficult to appraise considering that the results not well organized and do not tell the full story. References: References need to be formatted -Add hyperlinks for press releases and online articles.

Response:

The references have been formatted and hyperlinks have also been added to press releases and online articles, your input is greatly appreciated.

Reviewer: 2

Dr. Christina Carlander, Karolinska University Hospital

Comments to the Author:

This qualitative study performed in Zimbabwe and Malawi sought to deliver insights on the current state of awareness of viral suppression and treatment as prevention among PLHIV, influencers and healthcare providers. The study was well executed. The paper is well-written and easy to comprehend. It was interesting that so much was similar to the feelings/experiences that my patients (of whom a majority are migrants from sub-Saharan Africa) are sharing with me and co-workers.

Response:

Thank you so much for the compliment, it is also interesting that your colleagues have also shared similar experiences as our findings.

I have some comments.

- In the Introduction section (page 4, lines 47-57) four questions are posed which the study aimed to answer regarding awareness of treatment and viral suppression. Even though I appreciate that focus should be on the insights brought from the thematic analysis I would consider adding to the

Discussion a reflection on how well the study managed to answer the questions posed in the Introduction.

Response.

The aim of the study has been reworded (Page 4, lines 28-32) and it is now in line with the results and the discussion.

- How was the sample size decided upon? Was sampling done until redundancy in data was reached (as common in qualitative research)?

Response:

This was a multi-country research done during the COVID-19 peak period therefore sampling was also subjected to budgetary constraints. However the depth inquiry allowed to identify potential areas of interest.

- Consider including a little more data on PLHIV in Table 1. Time since HIV diagnosis? Time since start of ART? This would make it easier to assess transferability.

Response:

The selection criteria for PLHIV is well explained on Page 5, line 31-32 to Page 6 lines 1-4. The range of characteristics were wide and not easy to tabulate.

Reviewer: 3

Dr. Theodora Stroumpouki, Kingston University

Comments to the Author:

Dear authors,

Thank you for submitting a manuscript on the very important topic on the current state of awareness of viral suppression and treatment for PLHIV, their influencers, and healthcare providers. Please, consider the following comments constructively so that you can improve the content of this research paper.

The abstract sets the scene and gives to the reader a clear idea of the content of the paper. 'Design' and 'Setting and Participants' should be included in a broader section as 'Methodology'. The method of data analysis used should be also stated in the Abstract, prior to 'Results'.

Response:

*Thank you so much for your positive feedback in the effort of improving our manuscript quality. We have stated the method of analysis in our Abstract as suggested. **Page 2 Line 8-9.***

Introduction section: Could you please include a clear aim linked to the research questions?

Response:

*A clear aim of the study has been included (**Page 4, lines 28-32**) and we hope it now links well with the research questions.*

Methodology section: Could you please specify the type of qualitative design (line 7)? Is the design based on a specific theoretical framework? In section 2.3, could you please justify the reason you included only women? Is this due to gender inequality issue? If yes, could you please explain more and include evidence in the Introduction section that can provide the rationale for your choice? Could you please explain how trustworthiness has been achieved? Also, did the participants confirm the analysed results? How did this confirmation process occur? The data analysis section provides details on the process used to analyse the data. Was this process based on a specific model/step process?

Response:

*This qualitative research was the first part of broad research with 4 stages. The qualitative stage was conducted to build an understanding of unknown parts as explained in section 2.1, **Page 5 lines 10-15**. Young HIV positive women (18-35 years) were targeted as they have disproportionately low levels of viral suppression based on UNAIDS reports. This has been added in the introduction (. **Page 4, line 8-9**) and under section 2.3 Sample recruitment **Page 5, line 29-30**. Trustworthiness was achieved as explained on **Page 8 lines 27-32**. Data was also collected by trained moderators with vast experience in qualitative data collection. The data collection tools were also pretested (**Page 8, lines 15-25**).*

Ethical clearance: Were the participants able to withdraw from the study? How was confidentiality and anonymity maintained?

Response:

*The participants were able to withdraw from the study, stop the interview, request for the best time they felt the conversation could be continued as indicated on **Page 8, lines 21-25***

Results and Discussion were well presented. Findings are discussed in relation to the original research questions and sub-themes were well supported by narratives. Discussion includes secondary evidence that supports the findings. Citation 20 and 24 though are very old. Is there any specific reason for that? Could you find more current evidence?

Response:

Your feedback is very much appreciated. Citation 20 was authored in 2017 and Reference 24 in 2016, there was however an error in our reference system, we have since corrected it.

Strengths and Limitations: Do you think that the non-participation of male PLHIV could reveal different results? Could this be a limitation, too?

Response:

We totally agree with the reviewer, and are currently carrying out a similar study with male PLHIV, we are looking forward to the findings. We should be sharing the results by mid-2023.

Conclusions were well considered and some recommendations for further practice were provided.

The topic generates new knowledge on the essential components of HIV prevention, such as HIV education and awareness. Well done!

Response:

Your compliments are greatly appreciated.

Thank you for your consideration of this revised manuscript.

VERSION 2 – REVIEW

REVIEWER	Carlander, Christina Karolinska University Hospital
REVIEW RETURNED	02-Jan-2023

GENERAL COMMENTS	The review looks good. I have no further comments.
--

REVIEWER	Stroumpouki, Theodora Kingston University
REVIEW RETURNED	06-Feb-2023

GENERAL COMMENTS	Dear authors, Thank you very much for the time and effort placed in the amendments of your manuscript. I can see that you have considered and responded to the majority of the comments made by the reviewers. Well done! In the Data collection section though you do not explicitly document about participants' right to withdrawal. Although this is clearly mentioned in your comments to the Editor's letter, on the other hand it is not stated in the relevant section. Their right to withdraw from their study at any point before the commencement of the analysis, where participants cannot be identified due to anonymization, is very important to be considered and documented. Also, were you able to provide some support if participants felt stressed? As this topic is very sensitive and it can cause emotional distress, did you have any support in place for the participants? If yes, could you please state that in the data collection part?
--

	Glad to see that you have started a research study that addresses the same research questions for the males. All the best in your new journey!
--	--

VERSION 2 – AUTHOR RESPONSE

Reviewer: 2

Dr. Christina Carlander, Karolinska University Hospital

Comments to the Author:

The review looks good. I have no further comments.

Response: Thank you for your contribution.

Reviewer: 3

Dr. Theodora Stroumpouki, Kingston University

Comments to the Author:

Dear authors,

Thank you very much for the time and effort placed in the amendments of your manuscript. I can see that you have considered and responded to the majority of the comments made by the reviewers.

Well done!

Response: Thank you for the compliment and your efforts in improving the quality of our manuscript.

In the Data collection section though you do not explicitly document about participants' right to withdrawal. Although this is clearly mentioned in your comments to the Editor's letter, on the other hand it is not stated in the relevant section. Their right to withdraw from their study at any point before the commencement of the analysis, where participants cannot be identified due to anonymization, is very important to be considered and documented. Also, were you able to provide some support if participants felt stressed? As this topic is very sensitive and it can cause emotional distress, did you have any support in place for the participants? If yes, could you please state that in the data collection part?

Response: We totally agree with the reviewer that this was indeed a very sensitive. The data collection was done during the peak of COVID-19 and hence telephone interviews were conducted instead of face or face interviews. If at any point phone lines were interrupted or participants had to pause/stop the interview, the interviewers arranged to call back at a suitable time. (Page 8 lines 9-11). There were some participants who were referred for counselling and this arranged through their contacts sources

Glad to see that you have started a research study that addresses the same research questions for the males. All the best in your new journey!

Reviewer: 2

Competing interests of Reviewer: No competing interests

Reviewer: 3

Competing interests of Reviewer: None

Response: Your compliments are greatly appreciated.

Thank you for your consideration of this revised manuscript.